# Study on Co-Pyrolysis of Coal and Biomass and Process Simulation Optimization

**Biao Wang [1], Na Liu [1], Shanshan Wang [2], Xiaoxian Li [2], Rui Li [3,\*] and Yulong Wu [1,2,\*]**

1   School of Chemistry and Chemical Engineering, Xinjiang University, Ürümqi 830046, China; m18035193873@163.com (B.W.); ln@xju.edu.cn (N.L.)
2   Institute of Nuclear and New Energy Technology, Tsinghua University, Beijing 100084, China; wangshan624@163.com (S.W.); xx-li18@mails.tsinghua.edu.cn (X.L.)
3   MOE Engineering Center of Forestry Biomass Materials and Bioenergy, Beijing Forestry University, Beijing 100083, China
*   Correspondence: lirui@bjfu.edu.cn (R.L.); wylong@tsinghua.edu.cn (Y.W.)

**Abstract:** In this paper, the optimal process conditions for coal–biomass co-pyrolysis were obtained through pyrolysis experiments. The results show that under the condition of the pyrolysis temperature of 500 °C, the pyrolysis oil yield and positive synergistic effect reach the maximum, and the ratio of coal to biomass raw materials is 1:3. The effects of three loading methods (coal loading on biomass, biomass loading on coal, and coal–biomass mixing) on the distribution of simulated products of coal–biomass co-pyrolysis were constructed using Aspen Plus V11 software. The experimental results of pyrolysis carbon, pyrolysis oil, pyrolysis gas, and water under three different ratios are close to the simulation results, and the maximum error is 8%. This indicates that the model is dependent. This paper analyzes the economic situation in terms of investment in factory construction, raw material collection, product production, and product sales. The results show that when the processing scale is 9 tons $h^{-1}$, the pyrolysis plant can be profitable in the first year. This study provides basic data and the basis for the commercialization investment of coal–biomass co-pyrolysis technology.

**Keywords:** coal–biomass co-pyrolysis; optimum pyrolysis conditions; Aspen Plus; technical and economic analysis

## 1. Introduction

The characteristics of the energy structure in China are rich coal, poor oil, and little gas [1], with the proportion of coal in the energy structure of our country at 60.4%, placing it in the absolute dominant position. Although the coal reserves of China are relatively large, the quality of coal is not high. In general, the low-rank coal reserves account for more than half of the total coal reserves. Low-rank coal has high water content and volatile content, low ash content, and high activity. However, its thermal stability is poor, and its calorific value is low. However, the main utilization mode of low-rank coal in China is direct combustion, and about 90% of it is used for industrial power generation and boiler fuel. A large amount of energy is consumed in the combustion process, and the energy utilization efficiency is only about 35%. In addition, the combustion of low-rank coal is prone to cause serious environmental pollution and emits a large amount of greenhouse gases. Therefore, it is of great significance to rationally utilize coal resources, improve energy utilization efficiency, and reduce carbon emissions [2,3].

Biomass is a kind of clean and renewable energy, which can achieve zero carbon emission in the utilization process and reduce the emission of $SO_X$ and $NO_X$, thus reducing the pollution to the environment [4–6]. In recent years, the co-pyrolysis technology of biomass and coal has been widely studied by scholars, which is regarded as an effective way to improve the energy quality and utilization efficiency of coal and to solve the problem of environmental pollution [7–9]. Coal–biomass co-pyrolysis can not only reduce the use of

fossil energy and reduce greenhouse gas emissions but also improve the yield of pyrolysis oil with high calorific value in the pyrolysis products. In addition, the co-pyrolysis of coal and biomass can also solve the problem of the low energy density of biomass, which is not suitable for independent pyrolysis [10,11].

The co-pyrolysis process of biomass and coal is very complicated [12,13], which is difficult to be expressed with a series of accurate reaction equations. Moreover, traditional experimental methods cannot accurately calculate the reaction heat of pyrolysis process, which increases the difficulty of exploring the energy utilization efficiency of pyrolysis system. Aspen Plus V11, a large-scale process simulation software, provides a complete set of unit operation models for simulating various unit operation processes. Its physical property system is complete, including 1773 kinds of organic matter, 2450 kinds of inorganic matter, 3314 kinds of solid substances, and 900 kinds of water-soluble electrolytes [14,15]. In this paper, Aspen Plus V11 software was used to simulate the process of coal–biomass co-pyrolysis, and the simulation process of coal–biomass co-pyrolysis was built in combination with the experimental process. The influence of raw material feed amount and different filling methods on the industrial production cost of coal–biomass co-pyrolysis was analyzed. Through sensitivity analysis, the influence of the price changes of three kinds of products—pyrolytic carbon, pyrolytic oil, and pyrolysis gas—on the total return on investment was obtained, so as to provide a reference for improving the energy utilization efficiency of lignite in industrialization [16].

In this paper, the co-pyrolysis process of coal–biomass was studied using experimental methods, and the effects of pyrolysis temperature, filling method, and raw material ratio on the distribution and composition of co-pyrolysis products were investigated. The synergistic effect of coal and biomass was explored, and the optimum technological conditions for co-pyrolysis were obtained. The co-pyrolysis process of coal and biomass was simulated using Aspen Plus V11 software. The reliability of the model was verified by comparing the distribution of co-pyrolysis products under different raw material ratios with the experimental and simulated values. At last, through the economic analysis of the investment in plant construction, raw material collection, and product production and sales, this paper provides the basic data and basis for the commercialization of coal–biomass co-pyrolysis technology and the investment in plant construction.

## 2. Materials and Methods

### 2.1. Pyrolysis Experiment Design

#### 2.1.1. Raw Material

The biomass raw material is elm from Xinjiang, China. The biomass raw materials were crushed and screened into 0.15–0.30 mm particles. They were also dried in an oven at 105 °C for 12 h. The raw materials of Baishihu Coal from Hami of Xinjiang were crushed and sifted into fractions with particle sizes of 0.07–0.10 mm. They were also dried in an oven at 105 °C for 12 h, followed by the co-pyrolysis of coal and biomass. The proximate and ultimate analyses of coal and biomass are shown in Table 1.

**Table 1.** Proximate and ultimate analyses of samples (M: free water, A: ash content, V: volatile content, and FC: fixed carbon).

| Sample | Proximate Analysis (wt.%, ad) | | | | Ultimate Analysis (wt.%, daf) | | | | |
|---|---|---|---|---|---|---|---|---|---|
| | **M** | **A** | **V** | **FC** | **C** | **H** | **O** | **N** | **S** |
| Coal | 15.37 | 4.68 | 52.37 | 27.58 | 66.86 | 4.98 | 26.78 | 1.01 | 0.37 |
| Biomass | 3.26 | 2.26 | 75.16 | 19.32 | 47.90 | 5.59 | 46.12 | 0.39 | 0 |

#### 2.1.2. Experimental Apparatus and Process

The co-pyrolysis reaction of biomass and coal samples was carried out in a quartz tube fixed-bed reactor at 450 °C, 500 °C, 550 °C, 600 °C, 650 °C, and 700 °C. The loading

methods are divided into C#B, B#C, and Mix. C#B represents coal in the upper layer, B#C represents biomass in the upper layer, and Mix represents coal–biomass blend. The coal/biomass mixing ratio is 4:0, 3:1, 1:1, 1:3, and 0:4. The pyrolytic gas generated via pyrolysis is collected with a gas bag and analyzed using gas chromatography. The end of quartz tube is sealed with cotton for collecting pyrolytic oil. Pyrolytic oil was analyzed using GC-MS. Pyrolytic carbon was collected after cooling at the end of pyrolysis reaction. The reactor system is shown in Figure 1. The coal and biomass raw materials were weighed, respectively, according to the mixing ratio. The final total mass of 5.0 g [8,10,17] was placed in the center of the pyrolysis reactor. The carrier gas with a flow rate of 100 mL min$^{-1}$ N$_2$ (purity > 99.999%) flowed through the reactor system continuously. During the experiment, the temperature was heated from room temperature to reaction temperature at 10 °C min$^{-1}$ and maintained for 30 min. Finally, the temperature was lowered to 350 °C to finish the experiment. At the end of each experiment, the reactor was wiped with a sponge soaked in anhydrous ethanol and then burned and cleaned in the air before the next experiment was conducted.

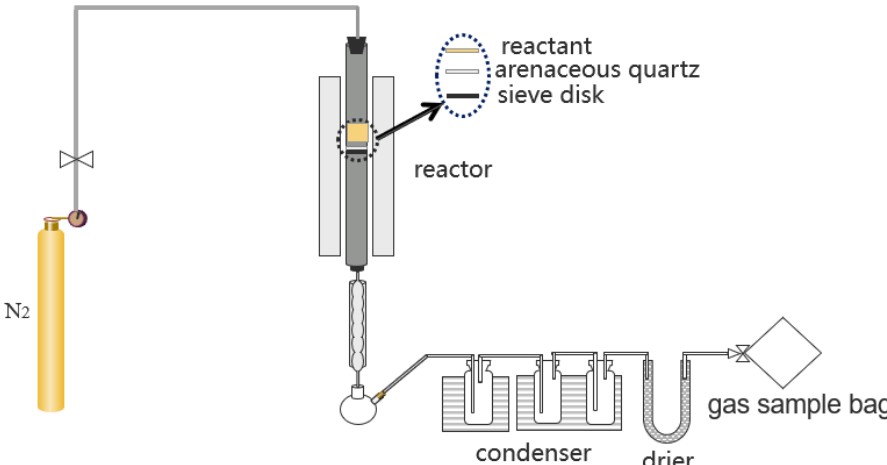

**Figure 1.** Schematic diagram of a fixed-bed reactor.

The purpose of coal–biomass pyrolysis is to explore the synergistic effect of their co-pyrolysis. In this section, the effects of raw material ratio, pyrolysis temperature, and filling method on the distribution and composition of co-pyrolysis products were studied, and the optimal process conditions were determined.

2.1.3. Product Analysis

The contents of eight main gases including CO, CH$_4$, CO$_2$, H$_2$, C$_2$H$_4$, C$_2$H$_6$, C$_3$H$_6$, and C$_3$H$_8$ in the gas-phase products were determined using gas chromatography. The weight of the reaction tube is weighed before the reaction, and the weight of the quartz tube is weighed after the reaction; the pyrolysis oil weight is obtained by reducing the difference. The pyrolysis oil was dissolved into toluene to measure its water content. A repeat experiment was conducted again, the pyrolysis oil was dissolved into dichloromethane, the water was filtered and removed. The error of the replicate experiment is required to be less than 5%. Pyrolysis oil is analyzed using a gas chromatograph–mass spectrometer (PerkinelmerClams500, GC-MS). The results of the GC-MS detection of pyrolysis oil were divided into phenol, pentene, pentane, ethanol, acetone, benzene, and methyl acetate.

The yield of pyrolysis gas and the volume concentration of eight main gases (CO, CH$_4$, CO, H$_2$, C$_2$H$_4$, C$_2$H$_6$, C$_3$H$_6$, and C$_3$H$_8$) are calculated using Formulas (1) and (2) below.

$$\text{Pyrolysis gas yield}: Y_i = \frac{V_i}{m_{feed}} \ (i = CO, \ CH_4, \ CO_2, \ H_2, \ C_2H_4, \ C_2H_6, \ C_3H_6, \ C_3H_8) \tag{1}$$

Volume concentration (Vol.%) : $V_i = \dfrac{V_i}{\sum Vi}$ (i = CO, CH$_4$, CO$_2$, H$_2$, C$_2$H$_4$, C$_2$H$_6$, C$_3$H$_6$, C$_3$H$_8$) (2)

In order to evaluate the conversion performance of pyrolytic oil, the yield of pyrolytic oil was calculated according to Formula (3).

$$\text{Tar yield} = \frac{m_{tar}}{m_{feed}}$$ (3)

Here, $m_{\text{tar}}$ and $m_{\text{feed}}$ represent the weight (g) of pyrolysis oil and raw materials.

### 2.2. Aspen Plus Process Simulation Design

2.2.1. Simulation Flow Description

The data source of the Aspen simulation process is obtained from the above experiments. The conditions in the simulation process are also set according to the above experimental conditions. According to the filling method, two different simulation processes of stratified pyrolysis and mixed pyrolysis were established. The simulation process consists of three parts: drying unit, pyrolysis unit, and separation unit. The main difference between the layered pyrolysis simulation process and the hybrid pyrolysis simulation process is the difference in the pyrolysis unit. The drying unit, for raw material drying and moisture removal, was only physical change. A stoichiometric reactor (RStoic) was selected for the drying of raw materials. The separation of dried coal and biomass raw materials and water was completed by a separator. In the mixed pyrolysis model, the dried coal and biomass were mixed using a mixer before entering the pyrolysis unit. Due to the complex process of the co-pyrolysis of coal–biomass, the reaction products (mainly including C, H$_2$, O$_2$, N$_2$, and S) were first normalized in the RYield [18,19] reactor according to the elemental analysis results, and then, pyrolysis was performed in the next RYield reactor. The pyrolysis products were divided into pyrolytic carbon, pyrolytic oil, pyrolysis gas, and water. According to the results of gas chromatographic analysis, pyrolysis gas consisted of CO, CH$_4$, CO$_2$, H$_2$, C$_2$H$_4$, C$_2$H$_6$, C$_3$H$_6$, and C$_3$H$_8$. According to GC-MS analysis results, pyrolytic oil was mainly composed of phenols; alkenes; alkanes; alcohols; ketones; benzenes; and esters. Therefore, it can be simplified into a mixture of seven model compounds, including phenol (C$_6$H$_6$O), pentene (C$_5$H$_{10}$), pentane (C$_5$H$_{12}$), ethanol (C$_2$H$_5$OH), acetone (CH$_3$COCH$_3$), benzene (C$_6$H$_6$), and methyl acetate (C$_3$H$_6$O$_2$). In addition, the pyrolytic carbon was mainly carbon. In the stratified pyrolysis model, coal or biomass was first pyrolyzed separately in an RYield reactor, and all the pyrolysis products were transferred to the next RYield reactor for mixed pyrolysis. In the separation unit, the pyrolytic products were first removed using the cyclone separator and then through the heat exchanger, the gas–liquid separation was carried out, and the pyrolysis gas was removed. The final liquid phase product was removed using the separator. Figure 2 shows the mixed pyrolysis model and coal–biomass pyrolysis layered model. Coal and biomass raw materials are unconventional components. The HCOALGEN model is used to calculate enthalpy value, and the DCOALIGT model is used to calculate density. In the pyrolysis process, the raw material is decomposed into conventional components with the yield reactor, and the final pyrolysis product is also represented with conventional components. The simulation flow chart is shown in Figure 2.

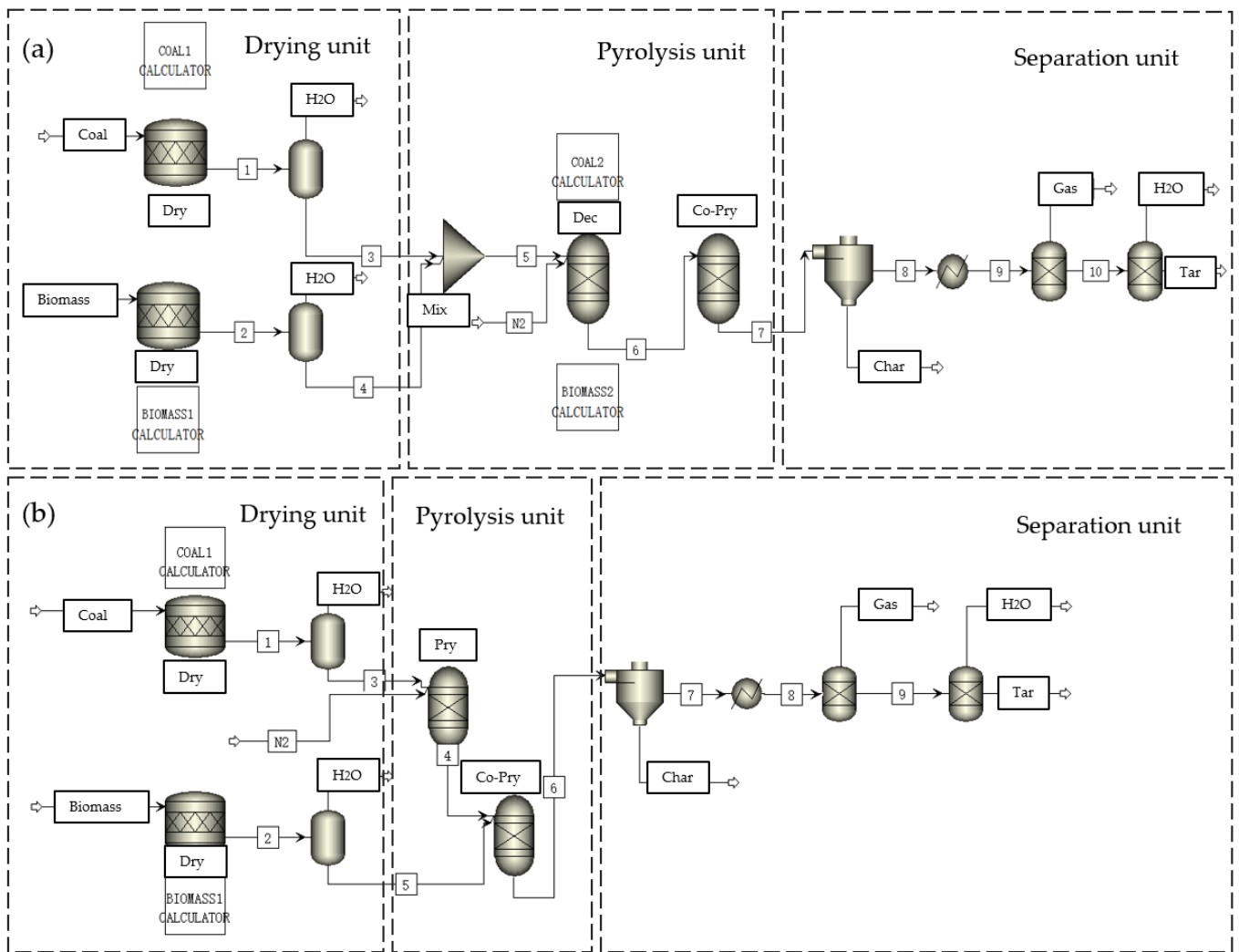

**Figure 2.** Simulation flow chart of co-pyrolysis of coal and biomass ((**a**) Coal–biomass mixed pyrolysis model and (**b**) stratified pyrolysis model of coal–biomass).

### 2.2.2. Model Building Assumptions

The following assumptions were made during the establishment of the co-pyrolysis model [14,20]:

(1)  The simulation system is a steady-state process;
(2)  Ash in coal and biomass is an inert component and does not participate in the pyrolysis simulation process;
(3)  Coal exists in the form of carbon, as does the pyrolytic carbon after the reaction;
(4)  It does not consider the loss of quality and energy transmitted between modules;
(5)  The pyrolysis process of the simulation process reaches equilibrium;
(6)  Gas-phase products with CO, $CH_4$, $CO_2$, $H_2$, $C_2H_4$, $C_2H_6$, $C_3H_6$, and $C_3H_8$, and these eight products were replaced.
(7)   According to the results of GC-MS, the pyrolysis oil was replaced with seven model compounds, including phenol ($C_6H_6O$), pentene ($C_5H_{10}$), pentane ($C_5H_{12}$), ethanol ($C_2H_5OH$), acetone ($CH_3COCH_3$), benzene ($C_6H_6$), and methyl acetate ($C_3H_6O_2$) [20,21].

### 2.2.3. Simulation Process Module and Function

The model modules and functions of coal–biomass co-pyrolysis process are shown in Table 2, which are represented in terms of modules and functions using drying units, pyrolysis units, and separation units.

**Table 2.** Coal and biomass co-pyrolysis model modules and functions.

| Operating Unit | Aspen Plus Module | Function |
|---|---|---|
| Drying unit | RStoic | Raw material drying |
| | Flash | Gas–solid separation (removing water) |
| Pyrolysis unit | RYield | The raw material decomposes into elemental substances |
| | RYield | Pyrolytic product generation |
| Separation unit | FSplit | Gas–solid separation |
| | Cooler | Oil–water condensation |
| | Sep | Pyrolysis gas separation |
| | Sep | Oil–water separation |

### 2.2.4. Aspen Plus Simulates the Pyrolysis Product Components

The co-pyrolysis products of coal–biomass can be divided into four parts: pyrolytic carbon, pyrolytic oil, pyrolysis gas, and water. The components and proportions of co-pyrolysis products are shown in Table 3.

**Table 3.** Components and proportions of simulated pyrolysis products.

| Compound | Molecular Formula | Mass Fraction (%) | | |
|---|---|---|---|---|
| | | (a) | (b) | (c) |
| liquid-phase product | - | - | - | - |
| phenol | $C_6H_6O$ | 9.71 | 11.3 | 11.4 |
| pentene | $C_5H_{10}$ | 0.71 | 0.83 | 0.87 |
| pentane | $C_5H_{12}$ | 1.17 | 1.36 | 1.31 |
| ethanol | $C_2H_5OH$ | 4.67 | 5.44 | 5.48 |
| acetone | $CH_3COCH_3$ | 3.69 | 4.29 | 4.25 |
| benzene | $C_6H_6$ | 2.34 | 2.73 | 2.77 |
| methyl acetate | $C_3H_6O_2$ | 0.9 | 1.05 | 1.01 |
| water | $H_2O$ | 22 | 21.4 | 20.7 |
| gaseous product | - | - | - | - |
| carbon monoxide | CO | 4.79 | 5.09 | 4.61 |
| methane | $CH_4$ | 1.37 | 1.45 | 1.32 |
| carbon dioxide | $CO_2$ | 9.96 | 10.54 | 9.59 |
| hydrogen | $H_2$ | 0.098 | 0.1 | 0.094 |
| ethylene | $C_2H_4$ | 0.098 | 0.1 | 0.094 |
| ethane | $C_2H_6$ | 0.098 | 0.1 | 0.094 |
| propylene | $C_3H_6$ | 0.098 | 0.1 | 0.094 |
| propane | $C_3H_8$ | 0.098 | 0.1 | 0.094 |
| solid-phase product | - | - | - | - |
| carbon | C | 38.2 | 37 | 36.31 |

(a) Composition and proportion of co-pyrolysis products obtained from mixed pyrolysis model. (b) Composition and proportion of co-pyrolysis products from stratified pyrolysis model of coal in top layer. (c) The composition and proportion of co-pyrolysis products from stratified pyrolysis model of coal in lower layer.

### 2.3. Technical and Economic Analysis

### 2.3.1. Economic Analysis Basis

Based on the experiment of the co-pyrolysis of coal and biomass and the simulation of Aspen Plus process, the economic analysis of the mixed pyrolysis and stratified pyrolysis process was carried out [22–24]. Coal and biomass are fed according to the optimal ratio of raw materials. All fees are adjusted based on 2023 USD exchange rates. Feedstock costs are based on local coal and biomass prices. Assuming 320 days a year, running 24 h a day, the estimated annual operating time is 7680 h, supposing the device has a lifespan of 20 years.

### 2.3.2. Cost of Capital

The capital cost of building the plant includes the cost of construction, equipment purchase, and equipment installation. Construction cost includes land cost and worker construction cost. The equipment purchase cost is mainly the equipment purchase cost of pyrolysis system. Most of the equipment cost comes from Aspen Plus V11, and part of the equipment cost comes from the literature [25–27]. The specific equipment cost is shown in Table 4. It is assumed that the construction time of the plant is 3 years, and the equipment life is 20 years. The key assumptions for the economic analysis are shown in Table 5, where the maximum processing scale is 10 t h$^{-1}$.

**Table 4.** Pyrolysis plant equipment cost [28].

| Unit | Equipment | Equipment Cost USD | Installed Cost USD | Quantity |
|---|---|---|---|---|
| Drying unit | Biomass and coal drying | 133,200 | 248,200 | 2 |
| | Flash vessel | 16,400 | 30,800 | 2 |
| Pyrolysis unit | Mixer | 37,000 | 69,000 | 1 |
| | Pyrolysis reactor | 497,800 | 630,200 | 2 |
| Separation unit | Cyclone | 13,000 | 37,000 | 1 |
| | Condensers | 43,500 | 236,300 | 1 |
| | Separator | 31,000 | 199,900 | 2 |
| | Storage tank | 52,000 | 78,000 | 2 |
| | Pipeline | 55,040 | 82,560 | |

**Table 5.** Total project investment factors [29].

| Component | Basis |
|---|---|
| Total Equipment Cost (TEC) | Equipment cost and Installed cost |
| Warehouse | 1.5% of TEC |
| Site Development | 9% of TEC |
| Total Installed Cost (TIC) | Sum of Above |
| Indirect Costs | |
| Field Expenses | 20% of TIC |
| Home Office and Construction Fee | 25% of TIC |
| Project Contingency | 3% of TIC |
| Total Capital Investment (TCI) | Sum of Above |
| Other Costs (Startup) | 10% of TCI |
| Total Project Investment | Sum of Above |

### 2.3.3. Operating Cost

Operating costs can be divided into variable and fixed operating costs. Variable operating costs mainly include power costs, cooling tower coolant, and raw material costs [15,21]. Electricity costs and cooling tower coolant were calculated according to the capital estimation tool in Aspen Plus V11. Fixed operating costs include labor, maintenance, overhead, taxes, and insurance. Fixed operating costs are shown in Table 6.

**Table 6.** Fixed operating costs [30].

| Positions Required | Number Required |
|---|---|
| Plant manager | 1 |
| Plant engineer | 1 |
| Maintenance supervisor | 1 |
| Lab manager | 1 |
| Shift supervisor | 3 |
| Maintenance tech | 9 |
| Shift operators | 33 |
| Yard employees | 18 |
| Clerks and secretaries | 3 |
| Total annual salaries | USD 1,600,000 |
| Maintenance | 3% of Total equipment cost |
| Insurance and taxes | 2% of Total installed cost |

Raw material costs include biomass and coal costs. The biomass is first cut down and transported to the factory and then crushed to the size required for pyrolysis. The cost of biomass raw materials includes the cost of cutting, transportation, afforestation, and crushing. Coal is purchased directly from the coal yard, transported to the factory, and then crushed to the size required for pyrolysis. The cost of coal includes the cost of buying, transporting, and crushing. The specific cost of raw materials is shown in Table 7.

**Table 7.** Components of feedstock cost [28].

| Material Cost | Components | USD t$^{-1}$ |
|---|---|---|
| Biomass cost | Cutting cost | 9.96 |
| | Skidding cost | 8.91 |
| | Crushing cost | 8.22 |
| | Road construction and infrastructure cost | 20.06 |
| | Afforestation cost | 30.64 |
| | Royalty/premium fee | 5.98 |
| | Loading, unloading, and transportation cost | 13.00 |
| | Delivered cost | 96.77 |
| Coal cost | Cost of purchase | 80.12 |
| | Loading, unloading, and transportation cost | 13.00 |
| | Crushing costs | 8.22 |
| | Delivered cost | 101.34 |

### 2.3.4. Product Sales

Pyrolytic oil, pyrolysis carbon, and pyrolysis gas in the pyrolysis products can be sold as products. Pyrolytic oil is sold with reference to the international oil price, which is about USD 569 t$^{-1}$. Pyrolytic carbon is marketed as a soil amendment for about USD 323 t$^{-1}$. The sale price of pyrolysis gas is about USD 590 t$^{-1}$ based on the natural gas price.

## 3. Results and Discussion

### 3.1. Analysis of Pyrolysis Experiment Results

In this section, the effects of pyrolysis temperature, raw material filling method, and raw material ratio on the product distribution and composition of the co-pyrolysis of coal and biomass were studied. On this basis, the optimal co-pyrolysis conditions were obtained. During the experiment, the three pyrolysis conditions of pyrolysis temperature, filling method, and raw material ratio were experimentally compared. It is easy to express in the text that when one of the optimum pyrolysis conditions is determined, the other two pyrolysis conditions are determined as the optimum pyrolysis conditions.

### 3.1.1. Choice of Filling Method

There are three different loading methods (C#B, B#C, and Mix) on the distribution of coal–biomass co-pyrolysis products. The best filling method is selected at the best temperature of 500 °C. The results are shown in Figure 3. The theoretical value is calculated from the results of a separate pyrolysis of coal–biomass. It can be seen from Figure 4 that the experimental value of pyrolysis oil for co-pyrolysis under the three loading methods is greater than the theoretical value. The B#C loading method has the highest output of pyrolytic oil and the lowest output of pyrolytic carbon, and pyrolysis oil is 7.6% higher than the theoretical value. The filling method has no effect on the yield of pyrolysis gas during co-pyrolysis. Biomass pyrolysis can form free radicals to react with coal. The formation of carbon residues in coal is inhibited [31]. In addition, the alkali and alkaline earth metals contained in biomass can promote the pyrolysis reaction of coal [13,32]. The pyrolysis reaction process in the experimental facility in this paper is from top to bottom, so the biomass has a better promotional effect on coal pyrolysis at the top.

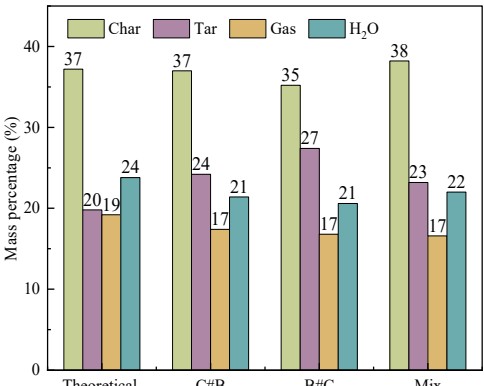

**Figure 3.** Component distribution of pyrolysis products using different packing methods.

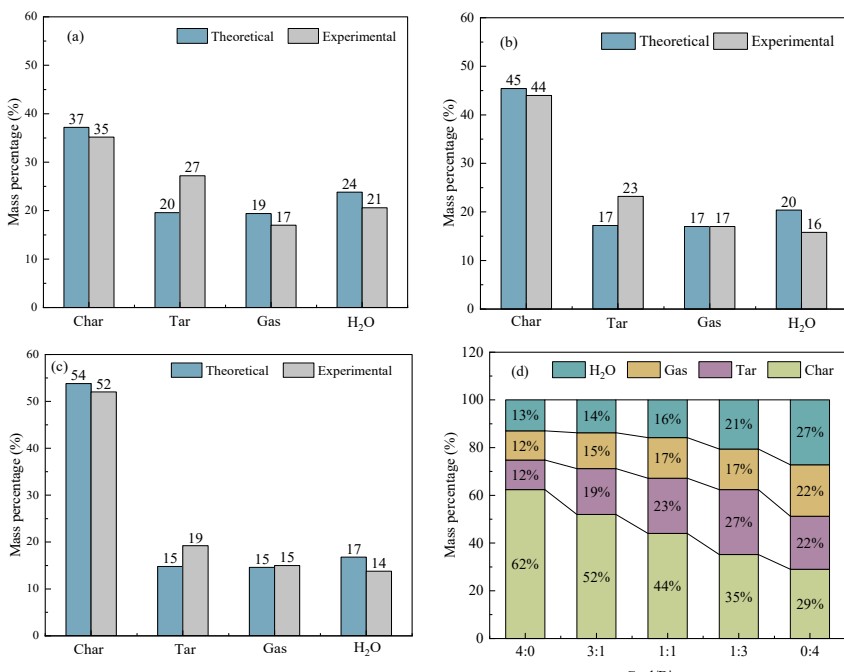

**Figure 4.** Theoretical values of different proportions, distribution of experimental products, and component distribution of pyrolysis products with different proportions. ((**a**) Coal to biomass ratio of 1:3, (**b**) coal/biomass ratio of 1:1, (**c**) coal/biomass ratio of 3:1, and (**d**) component distribution of pyrolysis products with different proportions).

### 3.1.2. Optimal Raw Material Ratio

The effects of different raw material ratios on the distribution of gas–liquid–solid three-phase products of coal–biomass co-pyrolysis were studied under the pyrolysis temperature of 500 °C and the loading method of B#C. The theoretical values were weighted by averaging the pyrolysis results of coal and biomass separately.

Figure 4a shows the comparison of theoretical and experimental values of different products when the mass ratio of coal to biomass is 1:3. The experimental value of pyrolytic carbon yield in pyrolysis products is less than the theoretical value, the experimental value of pyrolytic oil is significantly greater than the theoretical value, and the experimental value of gas phase product is slightly less than the theoretical value. The results show that the co-pyrolysis of coal and biomass at the ratio of 1:3 achieves the highest yield of pyrolysis oil [33].

Figure 4b shows the comparison of the theoretical and experimental values of different products when the mass ratio of coal to biomass is 1:1. The experimental value of pyrolytic carbon is less than the theoretical value, the experimental value of pyrolytic oil is greater than the theoretical value, and the experimental value of pyrolysis gas is basically equal to the theoretical value.

Figure 4c shows the comparison of theoretical and experimental values of different products when the mass ratio of coal and biomass is 3:1. The experimental value of pyrolytic carbon in pyrolysis products is slightly less than the theoretical value, the experimental value of pyrolytic oil is slightly greater than the theoretical value, and the experimental value of pyrolysis gas is basically equal to the theoretical value. It can be seen that the co-pyrolysis under the three ratios can promote the generation of pyrolysis oil. Three kinds of different ratios were greater than the theoretical value and experimental value of the water.

Figure 4d shows the distribution of co-pyrolysis products of coal and biomass at different ratios. Biomass contains a large number of volatile components, so the output of pyrolysis gas during biomass pyrolysis alone is greater than that during co-pyrolysis and coal pyrolysis separately. In the process of co-pyrolysis, the higher the proportion of biomass, the higher the yield of pyrolytic oil, while the higher the proportion of coal, the higher the yield of pyrolytic carbon, pyrolysis gas, and pyrolysis oil [34]. From the elemental analysis of coal and biomass, it can be seen that coal has a higher carbon content, and biomass has a higher oxygen content. Therefore, the product water increases with the increase in the biomass content during the co-pyrolysis of coal and biomass. Moreover, the content of pyrolysis carbon residue increases with the increase in coal content. Similarly, biomass has a higher volatile content. Therefore, the contents of pyrolysis oil and pyrolysis gas increase with the increase in the biomass content during the co-pyrolysis of coal and biomass [35]. Among the three ratios, when the ratio of coal and biomass of 1:3 showed the best synergistic effect, the yield of pyrolysis oil is the highest, and the pyrolysis carbon is the lowest [36].

### 3.1.3. Pyrolysis Temperature

The influence of pyrolysis temperature on the co-pyrolysis of coal and biomass in the range of 450–700 °C was studied under the conditions of 1:3 coal/biomass mixture ratio and B#C filling method. The results are shown in Figure 5. The results show that with the increase in temperature, the yield of pyrolytic carbon decreases gradually, and the Pyrolysis oil increases first and then decreases. Tar production reaches its maximum at 500 °C. With the increase in pyrolysis temperature, the production of pyrolysis gas increases gradually and reaches its maximum value at 700 °C. Since pyrolysis oil is the most important product in co-pyrolysis, 500 °C is chosen as the best pyrolysis temperature.

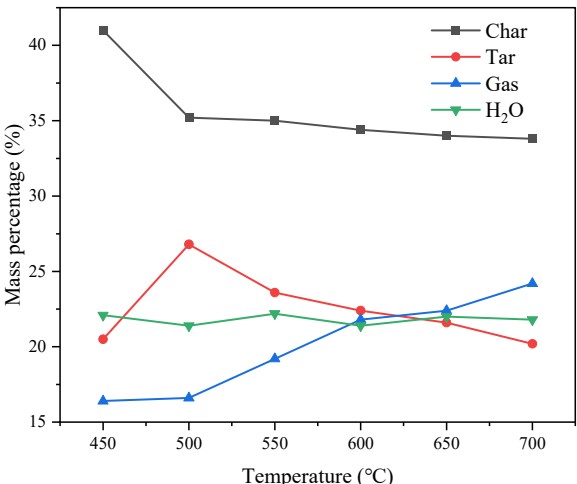

**Figure 5.** Product distribution with different pyrolysis temperatures. (Char: pyrolytic carbon; Tar: Pyrolysis oil; Gas: pyrolysis gas).

### 3.1.4. Analysis of Pyrolysis Products

Under the condition that the co-pyrolysis temperature is 500 °C and the loading method is B#C, the composition distribution of co-pyrolysis oil and pyrolysis gas at different ratios is studied. In Figure 6a, the figure is obtained by analyzing the GC-MS results. The pyrolysis oil components were divided into phenol from GC-MS; pentene; pentane; ethanol; acetone; Benzene; and methyl acetate. There is a comparison of the area occupied by the GS-MS results of each substance. When the coal/biomass ratio is 1:3, the contents of alcohols and ketones in the pyrolysis oil are the highest; when the coal/biomass ratio is 3:1, the contents of alkanes, lipids, and olefins in the pyrolysis oil are the highest. When the coal/biomass ratio is 1:1, the content of phenols in pyrolysis oil is the highest, and the content of other substances is between the other two ratios (that is, the ratios of coal and biomass are 3:1 and 1:3). Therefore, increasing the content of biomass in the mixed raw materials is conducive to increasing the contents of phenols, alcohols, and ketones in the pyrolysis oil. By contrast, the increase in coal is beneficial to increase the content of alkanes, lipids, and olefins in pyrolysis oil. Other groups of substances include acids, nitrogen, sulfur, and chlorine. Under the three ratios, the content of other substances is less than 3%, which proves that the quality of the pyrolysis oil from coal–biomass pyrolysis is much better.

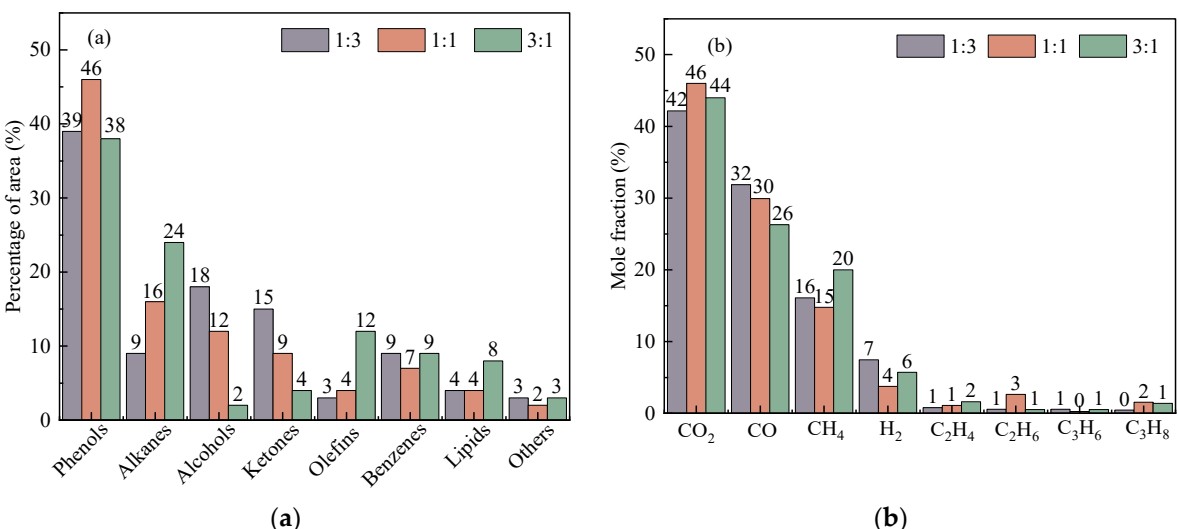

**Figure 6.** Pyrolysis product component analysis. ((**a**) Pyrolysis oil component analysis and (**b**) pyrolysis gas component analysis).

Figure 6b shows the result of the composition of the gas-phase products of co-pyrolysis under three different raw material ratios. The content of $CO_2$, CO, and $CH_4$ in pyrolysis gas is more than 80%, much higher than the other gas components. When the ratio of coal to biomass is 1:3, the content of CO and $H_2$ in gas-phase products is the highest. When the ratio of coal to biomass is 1:1, the content of $CO_2$; $C_2H_6$; and $C_3H_8$ in pyrolysis gas is the highest. When the ratio of coal to biomass is 3:1, the content of CH4 in pyrolysis is the highest. Due to the greenhouse effect of $CO_2$, the less $CO_2$ content in the pyrolysis gas products, the better. When the ratio of coal to biomass is 1:3, the content of $CO_2$ is the least, and the content of CO and $H_2$ is higher than that of the other two ratios, so the ratio is the best.

*3.2. Simulation Result Analysis*

### 3.2.1. Pyrolysis Oil Model Compound Analysis

This article classifies the GC-MS analysis results of pyrolysis oil and uses classified substances instead of whole pyrolysis oil for pyrolysis simulation work. These include phenol ($C_6H_6O$), pentene ($C_5H_{10}$), pentane ($C_5H_{12}$), ethanol ($C_2H_5OH$), acetone ($CH_3COCH_3$), benzene ($C_6H_6$), and methyl acetate ($C_3H_6O_2$). In order to verify the rationality of the model compound, the physical properties of the model compound and real pyrolysis oil were compared. Table 8 shows the comparison of the properties of real pyrolysis oil obtained in experiments with that constructed using model compounds, which are adopted in simulation. It can be seen that the total calorific value, net calorific value, and density of the model compound are slightly higher than that of pyrolysis oil, and differences in the physical properties such as heat capacity and viscosity are relatively larger. These differences mainly result from the small molecular weight of the model selected for simplification. Considering the fuel usage of the product, this simplification is considered reasonable as a whole.

**Table 8.** Comparison of physical properties of pyrolysis oil and model compounds.

| Property Comparison | Model Compounds | Pyrolysis Oil |
| :---: | :---: | :---: |
| Heat capacity (J/kg-K) | 2.17 | 1.78 |
| Viscosity (cp) | 0.97 | 0.64 |
| Total calorific value (MJ/kg) | 31.30 | 27.80 |
| Net calorific value (MJ/kg) | 29.25 | 26.11 |
| Density (kg/m$^3$) | 888.25 | 875.84 |

### 3.2.2. Pyrolysis Simulation Error Analysis

In this section, two models of mixed co-pyrolysis and stratified pyrolysis were established to explore the influence of three different loading methods on the distribution of the simulated products of coal–biomass co-pyrolysis. The results are shown in Figure 7. The experimental raw material amount is 5 g, and the simulated feeding amount is 2000 kg/h. Figure 7 compares the simulation results and experimental results at the pyrolysis temperature of coal and biomass at 500 °C, the ratio of coal and biomass at 3:1, and with the three loading modes. As can be seen from Figure 7, the experimental results of pyrolytic carbon, pyrolytic oil, pyrolysis gas, and water under three different ratios are close to the simulation results, in which the simulation relative error of coal on top is the largest, and the simulation relative error of uniform mixing is the smallest. The maximum error of the pyrolysis products is 8%, verifying that the model has good reliability.

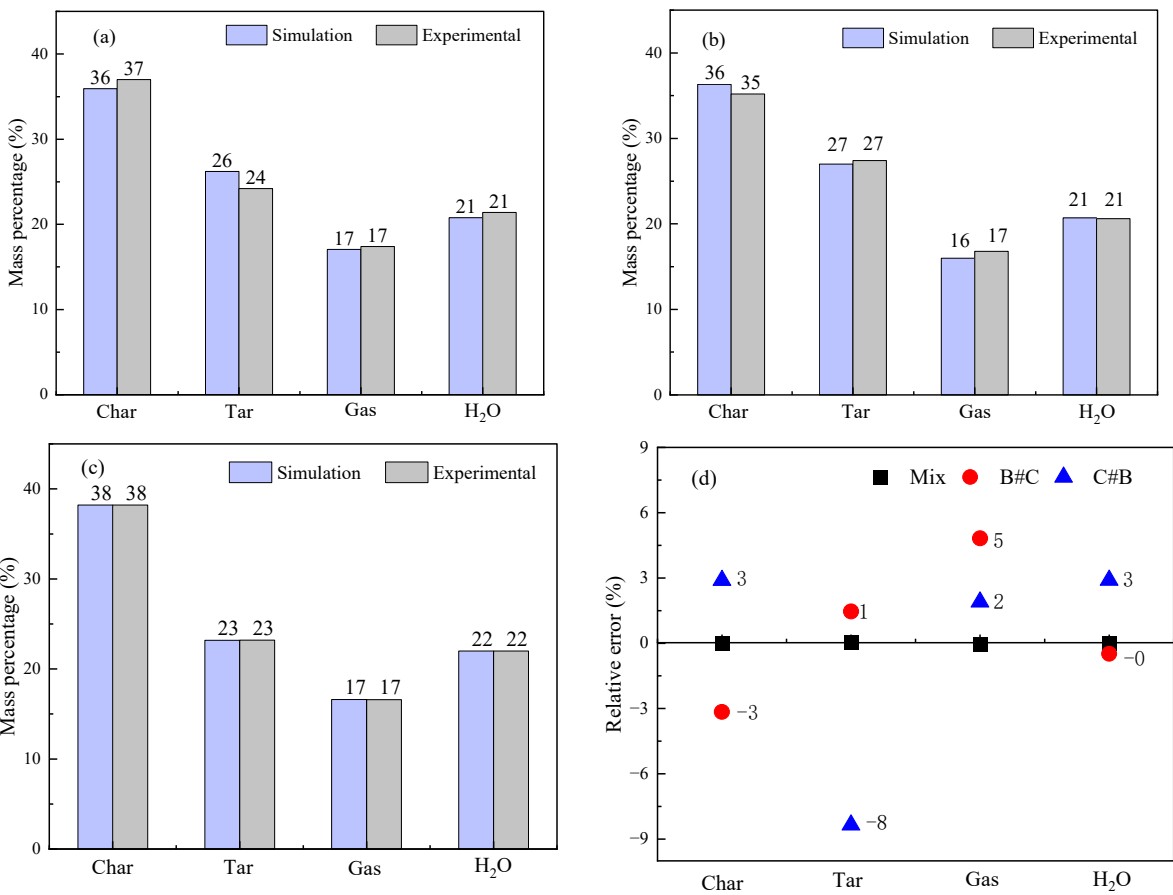

**Figure 7.** Product component distribution of simulated and experimental values and relative error. ((**a**) Filling method C#B, (**b**) filling method B#C, (**c**) filling method Mix, and (**d**) relative error).

### 3.3. Economic Analysis

Breakeven Analysis of Pyrolysis System

In this section, the economic analysis of the pyrolysis plant model built using three filling methods was carried out, respectively. Figure 8 shows the effect of processing scale on total profit. As can be seen from Figure 8, when the processing scale is 1 ton/h, the pyrolysis plants with the three loading methods have been in a state of loss during the 20-year running time. When the processing scale is 3 ton/h or above, the pyrolysis plant is in a profitable state. When the processing scale is 9 ton/h, the pyrolysis plant with three loading methods is the most profitable. Figure 8d shows the effect of the treatment scale on the total profit of different filler methods. The total profit is the sum of the profits of the pyrolysis plant for twenty years of operation. The total profit of the packing method C#B is higher than the total profit of the other two packing methods at different processing scales. The pyrolysis plant under the Mix packing method has the lowest total profit. From the perspective of profitability, the pyrolysis plant with packing method C#B is the most profitable. The larger the processing scale, the more profitable the plant. The optimal quantity of raw material feed is 9 t/h. The investment cost of coal–biomass co-pyrolysis is closely related to the production scale, and the larger the production scale, the less investment per unit production capacity as well as the higher the energy and material utilization rate.

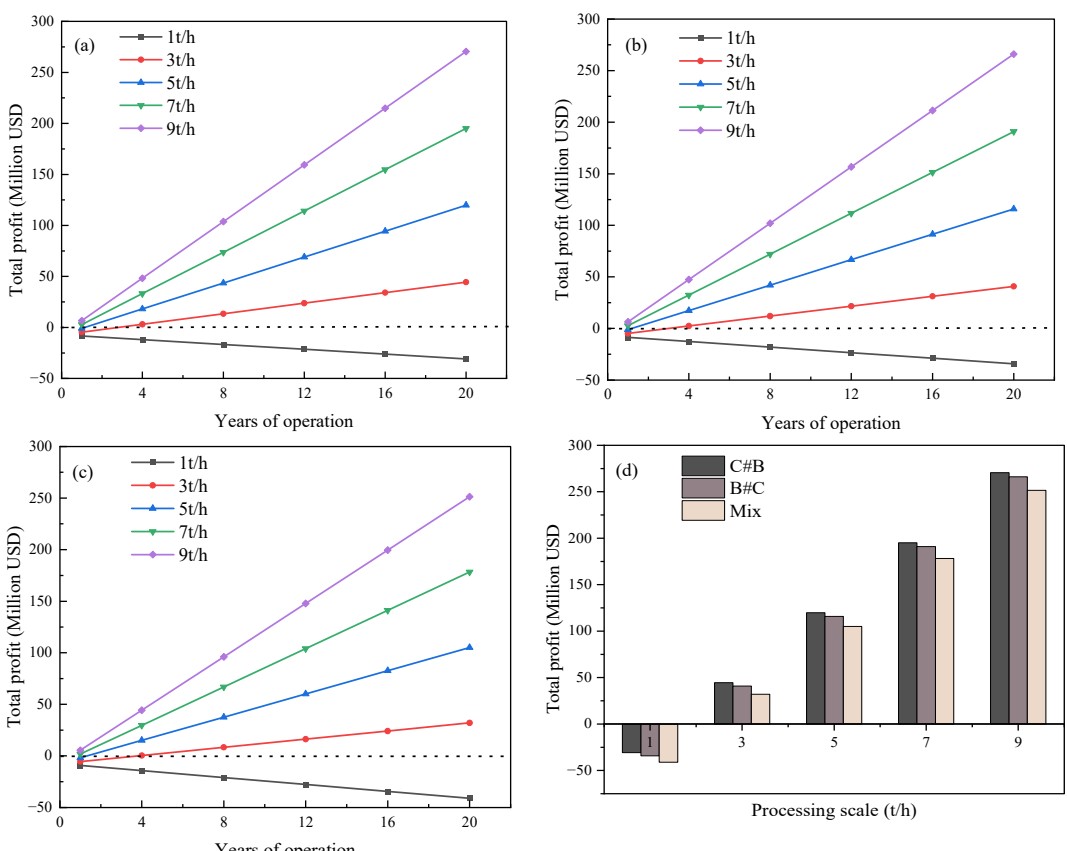

**Figure 8.** The impact of feed volume on total profit of packing method. ((**a**) Filling method C#B, (**b**) filling method B#C, (**c**) filling method Mix, and (**d**) comparison of the impact of the processing scale on the total profit of different filler methods).

## 4. Conclusions

In this paper, the coal–biomass co-pyrolysis experiment was first carried out. The best process conditions for coal–biomass co-pyrolysis were obtained. Then, the simulation process of the low-temperature pyrolysis of coal biomass was constructed using Aspen Plus V11 software. The research efficiency of coal–biomass co-pyrolysis energy conversion was improved, and the basic information of simulated pyrolysis energy consumption and capital cost of pyrolysis equipment was obtained. Finally, through the economic analysis of investment from plant construction, collection of raw materials and production products, and product sales, this paper provides the basic data and basis for the commercialization and investment of coal–biomass co-pyrolysis technology. The main research contents and conclusions of this paper are as follows:

(1) Firstly, the effects of pyrolysis temperature, filling method, and raw material ratio on the distribution and composition of co-pyrolysis products during the process of coal–biomass co-pyrolysis were studied. The components of pyrolysis oil and pyrolysis gas under relevant conditions were analyzed to explore the synergistic effect of coal and biomass in the process of co-pyrolysis, and the optimal technological conditions of coal–biomass co-pyrolysis were obtained. The results show that the maximum pyrolysis oil production is obtained under the optimal conditions of pyrolysis temperature at 500 °C, loading method B#C, and the raw material ratio of coal to biomass at 3:1, so the positive synergistic effect is the largest. The contents of phenols, alcohols, and ketones in pyrolysis oil were increased by increasing the proportion of biomass. The increase in coal increases the benzene, alkanes, lipids, and olefin in pyrolysis oil. This indicates that the quality of pyrolysis oil from coal–biomass pyrolysis is better. When the ratio of coal to biomass is 1:3, the content of $CO_2$ is the least, and the content of CO

and $H_2$ is higher than that of the other two ratios. Therefore, the quality of pyrolysis gas is better when the ratio of coal/biomass raw materials is 1:3;

(2) Aspen Plus V11 software was used to build the coal–biomass co-pyrolysis model for the two loading methods of stratified pyrolysis and mixed pyrolysis. The modeling scheme of each main unit in the system was determined, and the process simulation was carried out. According to the simulation results, the experimental values of pyrolytic carbon, pyrolytic oil, pyrolysis gas, and water under three different raw material ratios are close to the simulated values. The maximum error of the pyrolysis products is 8%, verifying that the model has good reliability.

(3) The economic analysis of investment and factory construction, raw material collection and product production, and product sales was carried out, and the impact of the raw material processing scale on total profit was explored. The results showed that when the processing scale is 1 ton/h, the pyrolysis plant is not economically efficient and is in a loss-making state. And when the processing scale is 9 ton/h, the pyrolysis plant can be profitable in the first year. This study provides the basic data and basis for the commercialization, investment, and construction of coal–biomass co-pyrolysis technology.

**Author Contributions:** B.W., writing—review and editing; N.L., methodology; S.W., visualization; X.L., validation; R.L., supervision; Y.W., project administration. All authors have read and agreed to the published version of the manuscript.

**Funding:** The authors are grateful for the financial support from the Huaneng Group science and technology research project (KTHT-U23YYJC01) and the National Natural Science Foundation of China (No. 21838006).

**Institutional Review Board Statement:** Not applicable.

**Informed Consent Statement:** Not applicable.

**Data Availability Statement:** Data are available upon request from the corresponding author.

**Conflicts of Interest:** The authors declare no conflict of interest.

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
