# Peer review of "Study on Co-Pyrolysis of Coal and Biomass and Process Simulation Optimization"

_sustainability, doi:10.3390/su152115412_

Round 1

Reviewer 1 Report

The quantities of material used in the experiments and in the simulations performed are very different.  A margin of error from simulation to experiment should be added, even if at first glance the differences are not more than 9%.

The conclusions presented at the end of the paper are consistent with the clarifications made along the way, but I don't know if the prices of raw materials and copyrolysis products are consistent with reality.

Reviewer 2 Report

Manuscript: sustainability-2551712-peer-review-v1

Title: Study on Co-pyrolysis of Coal and Biomass and Process Simulation Optimization

Reviews:

There are so many issues with the manuscript and the work done. First, there is no clear statement on what the research is trying to accomplish considering that co-pyrolysis of coal and biomass has been studied for a long time now and that several good papers have been published on the subject dating back to 2007 – a simple search in Web of Science will produce hundreds of papers on the topic. Was the study aimed at the kinetics? About the reaction steps? The results shown have yield values but many other yield values are already in the literature from several papers. What is this current work trying to bring to the literature of the subject?

If the work was about developing a simulation model of a process plant via AspenPlus software, the paper still is not clear on that approach. First, there is no clear explanation on how the empirical data (lab data) were adopted to the AspenPlus software. The software has a very strict rules on specifying the properties of chemicals and molecules so all atomic balances are met. For example this statement in lines 186-188 is not clear “(7) According to the results of GC-MS, the pyrolysis oil was replaced by seven model compounds, including phenol (C6H6O), pentene (C5H10), pentane (C5H12), ethanol (C2H5OH), acetone (CH3COCH3), benzene (C6H6) and methyl acetate (C3H6O2).” Pyrolysis oil is “oil” in liquid form, so how were these said model compounds behaving when used as model for pyrolysis oil? Was their aggregate behavior/properties similar to pyrolysis oil? This begs the question on how was this logic created for the statements in lines 153-159: “The pyrolysis products were divided into pyrolytic carbon, pyrolytic oil, pyrolysis gas and water. According to the results of gas chromatographic analysis, pyrolysis gas consisted of COCH4CO2H2C2H4C2H6C3H6 and C3H8. According to GC-MS analysis results, pyrolytic oil was mainly composed of phenolsalkenesalkanesalcoholsketonesbenzenes and esters. Therefore, it can be simplified into a mixture of seven model compounds, including phenol (C6H6O)pentene (C5H10)pentane (C5H12)ethanol (C2H5OH)acetone (CH3COCH3)benzene (C6H6) and methylacetate (C3H6O2).”?

Also, there is no clear explanation/show about the thermodynamic model computations used, i.e, the methods on how to estimate the reaction behavior and equation of state (pressure-volume-temperature) behavior of the fluids. Currently, there is no clear explanation on these. Even though there is a show of the process flow diagram (PFD) akin to the graphical interface of AspenPlus software, that diagram shown is not the model. An AspenPlus model is contained in an AspenPlus file and that file must also be submitted for review. That is fundamental even for an undergraduate course in chemical process design that uses the AspenPlus software – to submit the AspenPlus file. I work with AspenPlus software and I can check the AspenPlus file developed for this work – I have to check the AspenPlus – and the paper should upload that not just for review but also for publication. Please avoid using the statement “Data Availability Statement: Data is available upon request from the corresponding author.” If your intent is really to publish the work, then “publicize” the work including the data. If the authors still want to withhold such data including the AspenPlus files until someone requests, then at least submit those data for review by a reviewer to make sure that the modeling was correctly done.

The writing of the paper must also be improved. One indication that the writing is not of technical standards is the lengthy Abstract section. This is also apparent in the Conclusions section – why use numbers as the first terms in a paragraph and why write very lengthy recap of the contents of the manuscript? The Conclusions should briefly state how the findings of the work answer the main questions that the work are trying to answer – questions that are not clearly stated at this version of the paper. Please review how to write the various sections of a paper for a technical/scientific journal.

Please check the writing style of published peer-reviewed papers. There are many flaws in the writing of this paper.

Reviewer 3 Report

The article treats about the co-pyrolysis process of coal and biomass. The process was conducted in real conditions, and simulated. Boths were compared. The econoic analysis of solution implementation was also calculated and presented in the paper. The methods were well described. Perhaps, too much part is for methodology of economic analysis. However, I am aware that many assumptions must be taken to prepare the analysis of this kind. I am not a specialist in English, but in my opinion the text needs some grammar and language revisions, especially in Introduction and Methods... part. Some, re-organization of results description is needed. Details in the file attached.  

I am not a specialist in English, but in my opinion the text needs some grammar and language revisions, especially in Introduction and Methods... part.

Reviewer 4 Report

Dear Editor and Authors,

The manuscript (MS) is written on a current scientific and practical issue that expands the possibilities of using biomass together with and instead of coal. The MS obtained interesting results that need more in-depth scientific discussion. Please consider my recommendations and comments on MS improvement.

1) Please follow the classical structure and length of the abstract. Provide an introduction (urgency and significance of the research hypothesis), key results and conclusions (practical, commercial and environmental impacts).

2) The literature analysis on co-pyrolysis biomass and coal can be completed by reviewing relevant sources. For example, a recent publication in MDPI https://doi.org/10.3390/molecules28155708, co-pyrolysis of coking coals with cellulose, and a review of recent publications that extend the significance of your work https://doi.org/10.33271/mining17.02.001 , https://doi.org/10.4028/www.scientific.net/KEM.797.299

3) Why is the size chosen finer for coal than for biomass? What is the reason for the particle size? This is an important point, considering the influence of particle size on the pyrolysis process.

4) Why the moisture content of coal was so high (15.37 %) under such strict drying conditions?

5) The economic part of the manuscript could be significantly reduced. Much attention is focused on economic issues, while the manuscript has a scientific direction that requires significant improvement.

6) Fig. 3. Graphs should be provided with letter designations and captions in the figure title.

How can you explain the peak in water vapour at the 550оС?

7) The sentence on lines 270-272 should be rewritten for better clarity.

8) At temperatures up to 700oC coke is not formed (line 275). These temperatures correspond to semi-coke, but, probably, it is better to write as carbon residue from coal.

9) The statement on lines 276 and 277 is questionable and should be supported by references. The role of alkali and alkaline earth metals in pyrolysis reactions up to 700oC requires special consideration.

10) Fig. 4. It is recommended to provide numerical values on the diagrams for better clarity. Moreover, it would be desirable to indicate the standard deviation.

11) How does the finding that there are better synergies when the ratio of coal and biomass is 1:3 correspond with the results of other publications?

12) Fig. 5. It is recommended to indicate the standard deviation and the numerical values for the diagrams. How can you explain the increase in tar yield for mixtures when the amount of gas does not increase? Furthermore, why is there a decrease in water generation for mixtures? The MS lacks a scientific explanation of the results and comparison with other previously published work.

13) Fig. 6. For a complete analysis it is useful to provide results for the individual components.

14) In the Conclusions, general comments and the key results of the work should be given before listing the main findings.

The main shortcoming of the MS is the lack of scientific explanation of the results obtained.

Additionally, the MS should be proofread for technical and grammatical mistakes that occur.

Dear Editor,

The manuscript must be proofread. There are grammatical and technical mistakes. 

Round 2

Reviewer 2 Report

Manuscript: sustainability-2551712-peer-review-v2

Title: “Study on Co-pyrolysis of Coal and Biomass and Process Simulation Optimization”

Regarding authors’ response: “Kinetic analysis is not covered herein”. The issue with not performing kinetics modelling and using AspenPlus simulation is that the simulation is not modelling the kinetics of the reaction (because you did not develop kinetics model that can be used in the AspenPlus model). Using only Yield Reactors in AspenPlus does not correctly model the scaling of the reaction rates. Given that the paper includes Economic Analysis, the issue of scaling is crucial. A yield value from lab experiment does not linearly translate to the same yield value in pilot and large-scale systems. This is when kinetics (rate laws and reaction mechanisms) play roles in correct scaling of the model.

Regarding authors’ response: “In the research work, pyrolysis oil substances are classified according to the GC-MS results. Divided into phenol, pentene, pentane, ethanol, acetone, benzene and methyl acetate. These categories are then replaced with specific substances. The model compounds thus identified are reliable. In order to focus on tracking the changes of the main composition of the product, so as to establish the reactor model for subsequent practical application. Because model compounds are classified according to GC-MS results, their aggregate behavior and properties are similar to pyrolysis oil.” This is not a logical approach in concluding the MODEL results of approximating the properties of MIXED oil components via AspenPlus are good approximation of the experimental properties of the oil. We are questioning how good the AspenPlus MODEL approximations are of the oil properties measured. This question is asking for evidence that the AspenPlus MODEL of the oil behave in the same way as the experimental. The term ‘behave’ pertains to some properties check such as: Is the AspenPlus simulated oil still in the same phase as the experimental oil at specific experimental temperatures, pressures, etc.? Is the viscosity and other physical properties of the AspenPlus-simulated oil still close to the physical properties of the experimental oil?

Regarding authors’ response: “I provide a summary of the main results of the Aspen simulation process”. These screenshot results do not really mean anything during this review stage if the actual AspenPlus software file cannot be opened. I see an icon of AspenPlus at the end of the Author’s Response PDF document, but I do not see any file of AspenPlus models for review. This also makes me question the Data Availability statement that currently is written in the paper: “Data Availability Statement: Data is available upon request from the corresponding author.” I think it is about time that we, in the research community, should volunteer to make any data or software files be available without having to put ourselves (the authors of research papers) as gate-keepers of such information. In the end, a research work result that is used by many will highly likely result to faster developments in the research area.

Figure 7D: Do not use spline curves to connect data points when the factors are categorical. Spline curves have mathematical models behind them, and categorical data do not have such mathematical structures.

Figure 9: These graphs do not have any pertinent additional information delivered. It is intuitive that ROI increases as Profit increases. That is not even a research question. Remove Figure 9 and related discussion, perhaps the whole of Section 3.3.2 Sensitivity Analysis.

Minor language editing needed.

Author Response

Two aspen simulation streams of the original files were uploaded along with the manuscript.

Reviewer 4 Report

Dear Editor and Authors,

I would like to thank the authors for their detailed responses to the comments and corrections made to the manuscript (MS). The authors have improved the MS and I believe it can be considered by the editor for publication.

The current comment is that the references in the MS are inconsistently arranged.

Author Response

We thank you very much for giving us an opportunity to revise our manuscript, and thank you for your comments on our manuscript “Study on Co-pyrolysis of Coal and Biomass and Process Simulation Optimization”. We have carefully considered all comments and have made the revisions accordingly. We are now submitting the revised manuscript with descriptions and marks of suggested changes. We would like to express our great appreciation to you for your comments on our paper. Looking forward to hearing from you.

Thank you and best regards.

Yours sincerely,

Yulong Wu

Name: Yulong Wu

Response to the reviewer’s comments

Reviewer: 4

  1. The current comment is that the references in the MS are inconsistently arranged.

Response:

Thank you for your question about the typographic order of references. The revision of reference citations in the process of article revision does present the problem of typographical confusion. The reference order has been rearranged according to your requirements in the text. In my future work, I will pay more attention to detail.